# Prevalence of prelacteal feeding and associated risk factors in Indonesia: Evidence from the 2017 Indonesia Demographic Health Survey

**Lhuri D. Rahmartani**[1,2]*, **Claire Carson**[1●], **Maria A. Quigley**[1●]

**1** National Perinatal Epidemiology Unit, Nuffield Department of Population Health, University of Oxford, Oxford, United Kingdom, **2** Faculty of Public Health, Department of Epidemiology, Universitas Indonesia, Depok, Jawa Barat, Indonesia

● These authors contributed equally to this work.
* lhuri.rahmartani@dph.ox.ac.uk

## Abstract

### Background

Prelacteal feeding (PLF) is a recognised challenge to optimal breastfeeding but remains common in Indonesia. Meanwhile, PLF-related epidemiological research is limited, particularly in this setting. This study examines the prevalence and determinants of overall PLF as well as common PLF types (formula, other milk, and honey) in Indonesia.

### Methods

Data from 6127 mothers whose last child was ≤23-month-old were drawn from the 2017 Indonesia Demographic and Health Survey. Multivariable modified Poisson regression was used to measure the prevalence ratio (PR) for selected PLF risk factors. PLF was defined as anything to drink other than breast milk within three days after birth, before breastmilk flows. Additional analyses were performed on mothers who gave formula, other milk, and honey.

### Results

About 45% babies in Indonesia received PLF with formula being the most frequent (25%), followed by other milk (14%), plain water (5%), and honey (3%). Factors associated with higher prevalence of any PLF were higher wealth quintiles in rural area (PR 1.07; 95% CI 1.03–1.11 per increase in quintile), baby perceived to be small at birth (PR 1.23; 95% CI 1.12–1.35), caesarean deliveries at either public (PR 1.27; 95% CI 1.13–1.44) or private facilities (PR 1.15; 95% CI 1.01–1.31), and not having immediate skin-to-skin contact after birth (PR 1.32; 95% CI 1.23–1.42). PLF was less prevalent among mothers who gave birth to second/subsequent child (PR 0.82; 95% CI 0.76–0.88) and who had an antenatal card (PR 0.89; 95% CI 0.80–0.99). These patterns did not apply uniformly across all PLF types. For example, honey was more common among home births than deliveries at health facilities, but formula and other milk were more common among caesarean deliveries.

**Data Availability Statement:** Data are available from the The DHS Program database (https://dhsprogram.com/what-we-do/survey-survey-display-522.cfm).

**Funding:** This study was conducted as part of LDR's Doctor of Philosophy in Population Health at the University of Oxford. LDR receives the Jardine Scholarship (https://www.jardines.com) for her doctoral study at the Nuffield Department of Population Health, University of Oxford. LDR is supervised by CC, who is funded by a Career Development Award MR/L019671/1 from the Medical Research Council (https://mrc.ukri.org). The funders had no role in study design, data collection and analysis, decision to publish, or preparation of the manuscript.

**Competing interests:** The authors have declared that no competing interests exist.

## Conclusions

Mapping risk factors for PLF, especially by types, could help to design more targeted interventions to reduce PLF and improve breastfeeding practices in Indonesia.

## Introduction

Prelacteal feeding (PLF) describes any kind of feed other than breastmilk given to newborns before breastfeeding is established [1] and/or within several days after birth [2]. The types of feed vary widely across the world and may include infant formula milk, any other milk, water, sugar water, butter, honey, dates, and ghee [3–5]. While PLF may sometimes be medically indicated [6], it is also given for many other reasons. Some of the reported reasons are cultural or religious tradition [5,7], fear of newborn's hunger [8,9], perceived insufficiency of breastmilk [8,10,11], following advice from family or health providers [9,12], failure to initiate breastfeeding, and not rooming-in [13].

The World Health Organisation (WHO) warns against PLF because it can interfere with exclusive breastfeeding (EBF) [1], which is defined as giving nothing other than breastmilk for the first six months of life. PLF is considered either unnecessary [14], detrimental to breastfeeding [15], or even harmful to health as it is associated with an increased risk of infection [16] and avoidance of antibody-rich colostrum [17]. Hence, one of the WHO Ten Steps to Successful Breastfeeding is not providing food or fluids other than breastmilk for newborns, unless medically indicated (point no.6) [18,19].

Estimating the global prevalence of PLF practice is challenging because PLF-related studies mostly come from low and middle-income countries (LMICs), especially in Africa and Asia. A pooled study of Demographic Health Survey (DHS) datasets from 57 participating countries found that PLF is practiced by half of the mothers in LMICs, with Latin America having the lowest PLF prevalence (<40%) and Asia the highest (almost 60%) [20]. In Indonesia, a few studies have estimated the prevalence of PLF. For example, in 2017, prevalence of PLF ranged from 33% [21] to 44% [2]. Recent estimates from different regions in Indonesia have varied from 44% to 83% [8,9,12,13,22,23]. Several determinants of PLF practice in Indonesia have also been identified from these studies, including misperception, lack of knowledge, and cultural tradition, but due to small sample size and being restricted to particular region, these findings may not be generalizable to the wider population.

Breastfeeding is potentially one of the most cost-efficient and cost-effective public health interventions to decrease child mortality [24,25]. Being an emerging lower-middle income country with over quarter of a billion population [26,27], Indonesia could see real health and economic benefits from improved breastfeeding practices. Understanding PLF practice is important for optimal breastfeeding and achieving breastfeeding targets. In this study, we used data from the 2017 Indonesia DHS (IDHS) to estimate the prevalence of overall PLF and each type of feed. We also analysed the factors associated with overall PLF as well as factors associated with specific types of PLF namely formula, other milk, and honey.

## Methods

This is a cross-sectional study using secondary data drawn from the 2017 IDHS [2], which is available free-of-charge for public use at https://dhsprogram.com/what-we-do/survey/survey-display-522.cfm. [28] The DHS Program is a series of national-scale surveys which have been

conducted in over 90 countries since 1984, funded by the United States Agency for International Development (USAID) and implemented by ICF International in collaboration with local authorities–usually at ministerial or governmental level [29]. The first IDHS was completed in 1987 and, since 2002, it has been conducted every five years [28].

## Study population

As shown in Fig 1, the 2017 IDHS interviewed 49,627 women aged 15–49 from all 34 provinces in Indonesia, of whom 34,199 were mothers [2]. Questions on breastfeeding and PLF practice were asked of all mothers whose last child was aged 5-years or younger. However, the IDHS reports [30] and previous studies generally included estimates of breastfeeding indicators based on mothers whose last child was 0–23 month-old [3,20,31]. Therefore, to enable comparability with the existing literature as well as minimise recall bias [32], our study population was also limited to mothers whose last child was 0–23 months.

Mothers who never breastfed were excluded from the study population, as they could not, by definition, give a PLF. Finally, mothers who did not have complete data on all outcome and explanatory variables included in the analysis were excluded, and the final number included in the analysis was 6127. A complete case analysis was performed rather than imputation because the proportion of mothers missing any data was small (6.7%) [33].

## Outcome variables

As shown in Fig 1, the main study outcome was "any PLF". Those who answered "yes" to the prelacteal question were labelled as "any PLF" and compared with those who answered "no" (labelled as "no PLF"). Additional outcomes were based on the three types of PLF, which are "formula", "other milk", and "honey".

The questionnaire was asked in Indonesian [34] and when translated back to English, the PLF question was "In the first three days after delivery, before your milk began flowing regularly, was (NAME) given anything to drink other than breast milk?" [34]. This variable represents the umbrella term of all kinds of PLF given by mothers in the survey, and is comparable with the wider literature [3,4,20,35].

Mothers who answered "yes" to the PLF question were subsequently asked if they gave specific types of feed: infant formula milk, other milk, water, honey, glucose water, tea, rice water, juice, gripe water, sugar-salt solution, or others. Mothers answered "yes" or "no" to each type of feed; hence, there were mothers who gave two or more types of PLF.

For the additional outcomes, first we analysed "formula" (i.e. those who answered "yes" to infant formula milk) compared with "no PLF". Second, we analysed "other milk" (i.e. those who answered "yes" to any other milk) compared with "no PLF". Finally, we analysed "honey" (i.e. those who answered "yes" to honey) compared with "no PLF". Formula and other milk were chosen as additional outcomes because they were the two most common feeds. Honey was also selected because of its popularity in Indonesia [21] despite WHO's warning against honey being given to babies under one-year-old due to the risk of botulinum toxin [36].

## Explanatory variables

We followed the conceptual framework developed by Rollins et.al [37] in selecting the potential determinants of PLF and modified the classification based on the relevance and availability of survey variables. Eleven potential explanatory variables were divided into sociodemographic and birth-related factors. Sociodemographic variables of the mothers included maternal age (15–19, 20–24, 25–29, 30–34, and ≥35), level of education (none/primary, secondary, and higher), area of residence (urban and rural), and wealth quintile (Q1—Q5, where Q1 is the

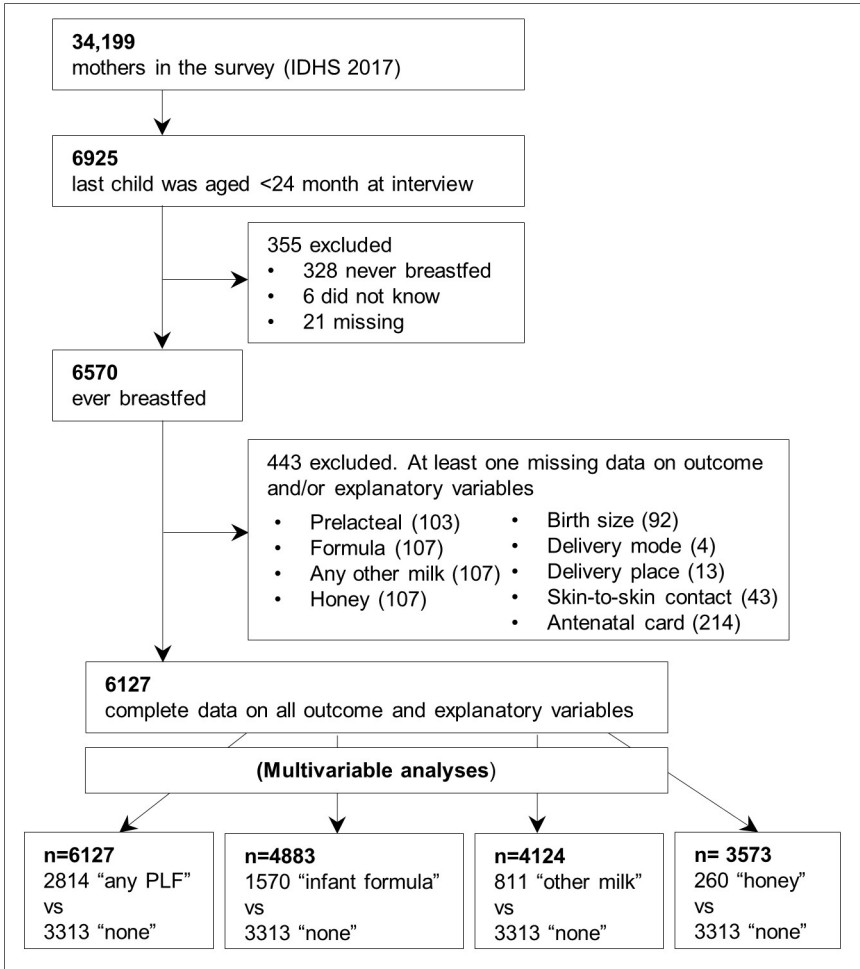

**Fig 1. Flow diagram of study population selection.**

least affluent group). The calculation of wealth quintile, which is the percent distribution of the de jure population by wealth quintiles and the Gini coefficient, is explained further on https://dhsprogram.com/Data/Guide-to-DHS-Statistics/Wealth_Quintiles.htm [30].

Birth-related variables included in this study were sex of the child (male and female), birth order (born first and born second/later), perceived birth size (mothers' perception of whether their babies were average, smaller, or larger), delivery place and mode, immediate skin-to-skin contact between mother and baby after birth, as well as possession of an antenatal card.

Delivery place and mode were combined into one factor because they were strongly associated with each other, since caesarean deliveries could not happen at home. Delivery place and mode comprised of home birth, vaginal delivery at public health facility, caesarean delivery at public health facility, vaginal delivery at private health facility, and caesarean delivery at private health facility. Possession of an antenatal card was defined as whether the mother had an antenatal card known as "Buku Kesehatan Ibu dan Anak", which translates to "Indonesian maternal and child health book". In this study, possession of an antenatal card was considered as a potential associated factor because it is designed as an educative intervention for pregnancy and birth preparedness. Antenatal cards include information on breastfeeding including PLF avoidance and colostrum benefit [38].

## Statistical analysis

The 2017 IDHS derived sampling weights so that the study sample is broadly representative of the population at the national level, provincial level, as well as urban-rural distribution.(40) All analyses were conducted using the 'svy' commands in Stata version 15 [39]. All proportions, prevalence ratios (PR), and statistical tests use weighted data while frequencies represent true counts (unweighted).

The prevalence of the study outcome (PLF) was considered high, so the associations between the outcome and the explanatory variables were assessed using modified Poisson regression, to minimise overestimation of the effect measure [40]. An initial univariable model was fitted to examine the association between each explanatory and outcome variable in the form of the crude PR. Multivariable analyses were conducted in three stages. First, multivariable models were adjusted for all sociodemographic factors. Next, they were adjusted for birth-related factors. Finally, both models were combined and factors that were not statistically significant (p>0.05) were removed from the model. Only explanatory factors that remained significant after adjusting for everything else (Wald p <0.05) were included in the final models.

There was evidence that wealth quintiles weren't distributed evenly across urban and rural areas [2], hence we fitted an interaction between wealth quintile (as a continuous variable) and area of residence in our models. The interaction was statistically significant for all outcomes (table not shown but available upon request); therefore, we present PRs for wealth quintile separately for urban and rural areas.

## Ethical review

Data from IDHS 2017 were obtained from procedures and questionnaires that comply with standard DHS surveys (https://dhsprogram.com/What-We-Do/Protecting-the-Privacy-of-DHS-Survey-Respondents.cfm). All protocols have been reviewed and approved by ICF Institutional Review Board (IRB) and an IRB in the host country, i.e. Indonesia in this case [41]. ICF IRB confirms that the survey conforms to the U.S. Department of Health and Human Services regulations for the protection of human subjects (45 CFR 46) [41].

# Results

## Study population

Of 6127 mothers in the study, the mean age was 29.5, 59.9% had secondary education, and 50.8% lived in rural areas (Table 1). One-third of the study population reported that the cohort member was their first baby. About 58% of the mothers perceived their baby to be of average birth size, while the rest perceived their baby to be smaller (11%) or larger (31%). The majority of the mothers delivered vaginally (14.7% at home, 27.8% at public health facilities, and 38.9% at private health facilities) while the rest had a caesarean delivery (7.0% at public health facilities and 11.7% at private health facilities). More than 65% of mothers had immediate skin-to-skin contact with their newborn.

## Prevalence and types of PLF in Indonesia

Table 2 shows the prevalence of PLF in general and by type in the IDHS 2017. Nearly 45% mothers gave their newborn something other than breastmilk in the first three days after birth. The most frequently reported PLF was formula, either exclusively (22.8%) or mixed with other feeds (2.5%), which together made up about half of all PLF. The next most common feed was other milk, which was given by 14.1% of mothers. Plain water and honey were third (5%) and fourth (3.4%), respectively. A very small percentage (less than 3%) of mothers gave other PLFs

**Table 1. Distribution of study population, overall and by type of residence.**

| Variables | Overall | | Urban | | Rural | |
|---|---|---|---|---|---|---|
| | **n** | **%** | **n** | **%** | **n** | **%** |
| | 6127 | 100.0 | 3047 | 100.0 | 3080 | 100.0 |
| **Sociodemographic factors** | | | | | | |
| **Maternal age** | | | | | | |
| 15–19 | 280 | 4.2 | 114 | 3.3 | 166 | 5.0 |
| 20–24 | 1210 | 20.1 | 530 | 17.9 | 680 | 22.3 |
| 25–29 | 1674 | 26.9 | 845 | 26.9 | 829 | 26.8 |
| 30–34 | 1566 | 25.9 | 815 | 27.3 | 751 | 24.6 |
| ≥35 | 1397 | 23.0 | 743 | 24.6 | 654 | 21.4 |
| **Education** | | | | | | |
| None/primary | 1371 | 22.8 | 442 | 14.9 | 929 | 30.4 |
| Secondary | 3515 | 59.9 | 1856 | 63.1 | 1659 | 56.9 |
| Higher | 1241 | 17.3 | 749 | 22.0 | 492 | 12.8 |
| **Wealth quintile** | | | | | | |
| Q1 (least affluent) | 1545 | 18.8 | 250 | 6.2 | 1295 | 30.9 |
| Q2 | 1244 | 20.6 | 473 | 13.1 | 771 | 27.8 |
| Q3 | 1142 | 20.1 | 673 | 21.8 | 469 | 18.5 |
| Q4 | 1124 | 20.8 | 771 | 27.2 | 353 | 14.7 |
| Q5 (most affluent) | 1072 | 19.7 | 880 | 31.7 | 192 | 8.1 |
| **Type of residence** | | | | | | |
| Urban | 3047 | 49.2 | 3047 | 100.0 | 0 | 0.0 |
| Rural | 3080 | 50.8 | 0 | 0.0 | 3080 | 100.0 |
| **Birth-related factors** | | | | | | |
| **Sex of the child** | | | | | | |
| Male | 3188 | 51.5 | 1586 | 51.0 | 1602 | 52.0 |
| Female | 2939 | 48.5 | 1461 | 49.0 | 1478 | 48.0 |
| **Birth order** | | | | | | |
| First | 1956 | 32.7 | 998 | 33.7 | 958 | 31.8 |
| Second/later | 4171 | 67.3 | 2049 | 66.3 | 2122 | 68.2 |
| **Perceived birth size** | | | | | | |
| Larger than average | 2038 | 31.3 | 946 | 29.4 | 1092 | 33.1 |
| Average | 3324 | 57.7 | 1755 | 60.3 | 1569 | 55.1 |
| Smaller than average | 765 | 11.1 | 346 | 10.4 | 419 | 11.8 |
| **Place and mode of delivery** | | | | | | |
| Home | 1202 | 14.7 | 273 | 6.0 | 929 | 23.2 |
| Public facility-vaginal delivery | 1889 | 27.8 | 722 | 20.2 | 1167 | 35.0 |
| Public facility-caesarean delivery | 519 | 7.0 | 300 | 8.1 | 219 | 5.9 |
| Private facility-vaginal delivery | 1915 | 38.9 | 1321 | 49.6 | 594 | 28.5 |
| Private facility-caesarean delivery | 602 | 11.7 | 431 | 16.1 | 171 | 7.4 |
| **Antenatal card** | | | | | | |
| No | 640 | 8.6 | 316 | 9.2 | 324 | 8.0 |
| Yes | 5487 | 91.4 | 2731 | 90.8 | 2756 | 92.0 |
| **Immediate skin-to-skin contact** | | | | | | |
| Yes | 3530 | 61.9 | 1902 | 65.7 | 1628 | 58.2 |
| No | 2597 | 38.1 | 1145 | 34.3 | 1452 | 41.8 |

**Table 2. Prevalence of PLF: Overall and by types of feed according to IDHS 2017.**

| PLF practice | n | % (weighted) |
|---|---|---|
| No PLF | 3313 | 55.3 |
| Any PLF | 2814 | 44.7 |
| Total | 6127 | 100.0 |
| **Types of PLF** | | |
| Infant formula milk | | |
| Infant formula only | 1417 | 22.8 |
| Infant formula + other PLFs | 153 | 2.5 |
| Total | 1570 | 25.3 |
| Other milk | | |
| Other milk only | 711 | 12.5 |
| Other milk + other PLFs | 100 | 1.7 |
| Total | 811 | 14.1 |
| Plain water | | |
| Plain water only | 138 | 2.0 |
| Plain water + other PLFs | 183 | 3.0 |
| Total | 321 | 5.0 |
| Honey | | |
| Honey only | 123 | 1.4 |
| Honey + other PLFs | 137 | 2.0 |
| Total | 260 | 3.4 |
| Glucose water | | |
| Glucose water only | 54 | 0.5 |
| Glucose water + other PLFs | 52 | 0.6 |
| Total | 106 | 1.1 |
| Tea | | |
| Tea only | 5 | 0.0 |
| Tea + other PLFs | 26 | 0.4 |
| Total | 31 | 0.4 |
| Rice water | | |
| Rice water only | 8 | 0.1 |
| Rice water + other PLFs | 9 | 0.2 |
| Total | 17 | 0.2 |
| Juice | | |
| Juice only | 4 | 0.0 |
| Juice + other PLFs | 12 | 0.2 |
| Total | 16 | 0.2 |
| Gripe | | |
| Gripe water only | 1 | 0.0 |
| Gripe water + other PLFs | 0 | 0.0 |
| Total | 1 | 0.0 |
| Unspecified others | 64 | 0.9 |

*) The denominator was overall study population i.e. mothers whose last child was 0 to 23 month-old at the time of interview, reported as having ever breastfed, and had complete data on outcome and explanatory variables (n = 6127). All % are weighted, to account for design effects.

such as glucose water, tea, juice, rice water, gripe water, and other feeds that were not specified in the IDHS questionnaire.

## Factors associated with any PLF

Table 3 (n = 6127) shows that the prevalence of any PLF did not differ significantly by maternal age, education, wealth quintile, or by the sex of the baby. However, giving any PLF was more common among mothers who lived in rural areas, had their first child, perceived their babies as smaller than average, had a caesarean delivery at public facilities, and did not have an antenatal card.

Table 3 shows the final models after mutual adjustment for all factors. The prevalence of any PLF was higher among mothers who perceived their baby to be smaller (aPR 1.23; 95% CI1.12–1.35) than among mothers who perceived their baby to be of average size. Compared to home births, having a caesarean delivery either in public facilities (aPR 1.27; 95% CI 1.13–1.44) or private facilities (aPR 1.15; 95% CI 1.01–1.31) was associated with a higher PLF prevalence. Conversely, a lower prevalence for any PLF was found in mothers giving birth to second/later child (aPR 0.82; 95% CI 0.76–0.88) and having an antenatal card (aPR 0.89; 95% CI 0.80–0.99). Wealth quintile was only statistically significant in rural areas but not in urban areas, with an increase of 1.07 (95% CI 1.03–1.11) for every increase in wealth quintile. Compared to having an immediate skin-to-skin contact after birth, not having it was associated with a higher prevalence of any PLF (aPR 1.32; 95% CI 1.23–1.42).

## Factors associated with formula

In Table 4, formula was more common among mothers who gave birth for the first time, perceived their baby to be smaller, and had a caesarean delivery (in either public or private facilities). For the Poisson regression models, we restricted the study population to those who gave formula and those who gave no PLF (n = 4883, Table 4).

In the final model (Table 4), positive associations were found among mothers with a perceived smaller baby (aPR 1.37; 95% CI 1.18–1.59) in comparison to those who perceived their baby as average size. Compared to home births, formula was more common among caesarean deliveries either in public (aPR 1.54; 95% CI 1.27–1.87) or private facilities (aPR 1.37; 95% CI 1.12–1.68). Similar to any PLF, mothers who gave birth to their second/later child were less likely to give formula (aPR 0.77; 95% CI 0.69–0.86) than were first-time mothers. Wealth quintile was statistically significant in rural areas but not in urban areas, with an increase of 1.12 (95% CI 1.06–1.19) for every increase in wealth quintile. Compared to having immediate skin-to-skin contact after birth, not having it was associated with a higher prevalence of prelacteal formula (aPR 1.44; 95% CI 1.29–1.60).

## Factors associated with other milk

Birth-related factors associated with other milk were broadly similar to those associated with formula (Table 4, n = 4124). After adjusting for all factors (Table 4), mothers with perceived smaller babies were more likely to give other milk (aPR 1.42; 95% CI 1.15–1.75) compared to mothers with babies of perceived average size. In comparison to home births, PRs were higher among vaginal deliveries in private facilities (aPR 1.43; 95% CI 1.09–1.86) and caesarean deliveries in either public (aPR 2.26; 95% CI 1.70–3.00) or private facilities (aPR 1.77; 95% CI 1.31–2.41). Similar to any PLF, having a second/later birth and having an antenatal card were associated with a lower prevalence of other milk (aPR 0.77; 95% CI 0.66–0.89 and aPR 0.73; 95% CI 0.59–0.91, respectively). Wealth quintile was statistically significant in rural areas but not in urban areas, with an increase of 1.14 (95% CI 1.04–1.124) for every increase in wealth quintile.

**Table 3. Prevalence of any PLF by sociodemographic and birth-related factors and PR (n = 6127).**

| Variables | No PLF | | Any PLF | | cPR (95% CI) | aPR (95%) |
|---|---|---|---|---|---|---|
| | n = 3313 | % | n = 2814 | % | | |
| **Sociodemographic factors** | | | | | | |
| **Maternal age** | | | | | | |
| 15–19 | 159 | 55.9 | 121 | 44.1 | 0.99 (0.81–1.21) | |
| 20–24 | 618 | 51.5 | 592 | 48.5 | 1.09 (0.98–1.20) | |
| 25–29 | 913 | 55.3 | 761 | 44.7 | Ref | |
| 30–34 | 869 | 56.6 | 697 | 43.4 | 0.97 (0.88–1.08) | |
| ≥35 | 754 | 57.1 | 643 | 42.9 | 0.96 (0.87–1.06) | |
| **Education** | | | | | | |
| None/primary | 783 | 57.5 | 588 | 42.5 | 0.94 (0.86–1.03) | |
| Secondary | 1906 | 54.9 | 1609 | 45.1 | ref | |
| Higher | 624 | 53.8 | 617 | 46.2 | 1.03 (0.94–1.12) | |
| **Wealth quintile** | | | | | | |
| Urban | | | | | | |
| Q1 (least affluent) | 134 | 51.1 | 116 | 49.0 | ref | ref |
| Q2 | 252 | 58.7 | 221 | 41.3 | 0.99 (0.95–1.03)* | 0.98(0.94–1.03)* |
| Q3 | 366 | 57.8 | 307 | 42.2 | | |
| Q4 | 422 | 58.4 | 349 | 41.6 | | |
| Q5 (most affluent) | 480 | 57.0 | 400 | 43.1 | | |
| Rural | | | | | | |
| Q1 (least affluent) | 779 | 59.9 | 516 | 40.1 | ref | ref |
| Q2 | 398 | 53.8 | 373 | 46.2 | 1.07 (1.03–1.12)* | 1.07 (1.03–1.11)* |
| Q3 | 233 | 49.1 | 236 | 50.9 | | |
| Q4 | 158 | 43.2 | 195 | 56.8 | | |
| Q5 (most affluent) | 91 | 54.3 | 101 | 45.7 | | |
| **Type of residence** | | | | | | |
| Urban | 1654 | 57.4 | 1393 | 42.6 | ref | |
| Rural | 1659 | 53.3 | 1421 | 46.7 | **1.1 (1.01–1.18)** | |
| **Birth-related factors** | | | | | | |
| **Sex of the child** | | | | | | |
| Male | 1727 | 55.6 | 1461 | 44.4 | ref | |
| Female | 1586 | 55.0 | 1353 | 45.0 | 1.01 (0.94–1.09) | |
| **Birth order** | | | | | | |
| First | 931 | 48.6 | 1025 | 51.4 | ref | ref |
| Second/later | 2382 | 58.6 | 1789 | 41.4 | **0.81 (0.75–0.87)** | **0.82 (0.76–0.88)** |
| **Perceived birth size** | | | | | | |
| Larger than average | 1116 | 55.1 | 922 | 44.9 | 1.05 (0.97–1.14) | 1.04 (0.97–1.13) |
| Average | 1836 | 57.3 | 1488 | 42.7 | ref | ref |
| Smaller than average | 361 | 45.5 | 404 | 54.5 | **1.28 (1.16–1.40)** | **1.23 (1.12–1.35)** |
| **Place and mode of delivery** | | | | | | |
| Home | 654 | 53.2 | 548 | 46.8 | ref | ref |
| Public facility-vaginal delivery | 1151 | 60.8 | 738 | 39.2 | **0.84 (0.75–0.94)** | 0.91 (0.82–1.02) |
| Public facility-caesarean delivery | 184 | 38.5 | 335 | 61.5 | **1.32 (1.17–1.48)** | **1.27 (1.13–1.44)** |
| Private facility-vaginal delivery | 1065 | 57.9 | 850 | 42.1 | 0.9 (0.81–1.00) | 1.00 (0.89–1.12) |
| Private facility-caesarean delivery | 259 | 46.4 | 343 | 53.6 | **1.15 (1.01–1.30)** | **1.15 (1.01–1.31)** |
| **Antenatal card** | | | | | | |
| No | 308 | 49.6 | 332 | 50.4 | ref | ref |

*(Continued)*

**Table 3.** (Continued)

| Variables | No PLF | | Any PLF | | cPR (95% CI) | aPR (95%) |
|---|---|---|---|---|---|---|
| | n = 3313 | % | n = 2814 | % | | |
| Yes | 3005 | 55.9 | 2482 | 44.2 | **0.88 (0.79–0.97)** | **0.89 (0.80–0.99)** |
| **Immediate skin-to-skin contact** | | | | | | |
| Yes | 1727 | 61.4 | 1461 | 38.6 | ref | ref |
| No | 1586 | 45.4 | 1353 | 54.6 | **1.42 (1.32–1.52)** | **1.32 (1.23–1.42)** |

(*) = wealth quintile as continuous variable; bold figure = statistically significant; n = unweighted count; % = weighted percent.

Compared to having immediate skin-to-skin contact after birth, not having it was associated with a higher prevalence (aPR 1.49; 95% CI 1.27–1.75).

### Factors associated with honey

The prevalence of honey showed different patterns across many of the factors compared to the other two feeds. When restricting the study population to those who gave honey and those who gave no PLF (n = 3573, Table 4), honey was most commonly given as PLF by mothers who were younger, had lower education, came from lower health quintiles in urban areas, and had a home birth. However, after adjusting for all factors, statistically significant associations were only found in birth order and place of delivery (Table 4). Higher birth order had a lower prevalence for honey (aPR 0.60; 95% CI 0.45–0.81), as was the case with any PLF and the other feeds. However, in contrast to the other feeds, home births had the highest prevalence while caesarean deliveries at private facilities had the lowest prevalence for honey (aPR 0.10; 95% CI 0.04–0.23). Wealth quintile was not a statistically significant factor in either urban or rural areas. Similar to the other outcomes, not having immediate skin-to-skin contact is associated with a higher prevalence of prelacteal honey (aPR 1.52; 95% CI 1.11–2.09).

We also analysed delivery place and mode separately (not shown but available upon request) and found that home births increased the prevalence by almost 5 times compared to delivery at private health facilities (aPR 4.76; 95% CI 2.91–7.78). When analysed without delivery place (home, public or private facility), mode of delivery (vaginal or caesarean) was not a statistically significant determinant for honey.

## Discussion

Our study found that in 2017 PLF was practised by approximately 45% of mothers in Indonesia, with formula being the most common feed. Higher wealth quintiles in rural areas, first birth, smaller babies, caesarean delivery, and not having an antenatal card were positively associated with giving any PLF. However, these patterns did not apply uniformly across different types of PLF. For example, although caesarean deliveries increased the risk for both formula and other milk, honey was more prevalent among home births.

### PLF prevalence and types

The latest estimate of PLF prevalence at 45% may be part of an ongoing decline in PLF in Indonesia as it was 64.6% in 2007 [42] and 60.3% in 2012 [43]. However, Indonesia remains behind other Southeast Asian countries such as Timor-Leste, the Philippines, and Myanmar, whose most recent PLF prevalence was 18%, 24%, and 20%, respectively [44–46]. Importantly, the higher prevalence in Indonesia is despite the Indonesian questionnaire using a stricter

**Table 4. Prevalence of PLF types by sociodemographic and birth-related factors and PR.**

| | Formula (n,%) | | cPR (95% CI) | aPR (95%) | Other milk (n, %) | | cPR (95% CI) | aPR (95%) | Honey (n, %) | | cPR (95% CI) | aPR (95%) |
|---|---|---|---|---|---|---|---|---|---|---|---|---|
| **Sociodemographic factors** | | | | | | | | | | | | |
| **Maternal age** | | | | | | | | | | | | |
| 15–19 | 70 | 32.6 | 1.03 (0.77–1.38) | | 27 | 15.6 | 0.76 (0.48–1.19) | | 15 | 10.1 | 1.51 (0.77–2.95) | |
| 20–24 | 313 | 34.6 | 1.09 (0.94–1.26) | | 162 | 21.0 | 1.02 (0.82–1.27) | | 57 | 7.5 | 1.12 (0.72–1.75) | |
| 25–29 | 433 | 31.7 | ref | | 221 | 20.6 | ref | | 79 | 6.7 | ref | |
| 30–34 | 382 | 29.4 | 0.93 (0.80–1.07) | | 216 | 21.5 | 1.04 (0.84–1.30) | | 67 | 5.0 | 0.76 (0.50–1.14) | |
| ≥35 | 372 | 30.3 | 0.96 (0.83–1.11) | | 185 | 19.1 | 0.92 (0.74–1.15) | | 42 | 3.7 | 0.55 (0.35–0.88) | |
| **Education** | | | | | | | | | | | | |
| None/primary | 281 | 27.3 | 0.84 (0.74–0.97) | | 152 | 17.5 | 0.85 (0.69–1.04) | | 73 | 7.8 | 1.49 (1.03–2.16) | |
| Secondary | 919 | 32.4 | Ref | | 473 | 20.7 | ref | | 136 | 5.2 | ref | |
| Higher | 370 | 33.3 | 1.03 (0.91–1.17) | | 186 | 23.1 | 1.12 (0.92–1.36) | | 51 | 5.5 | 1.05 (0.68–1.63) | |
| **Wealth quintile-urban** | | | | | | | | | | | | |
| Q1 (least affluent) | 62 | 34.5 | ref | ref | 26 | 17.8 | ref | ref | 18 | 12.0 | ref | ref |
| Q2 | 122 | 26.7 | 1 (0.94–1.06)* | 0.98 (0.92–1.04)* | 69 | 20.1 | 1.06 (0.97–1.16)* | 1.02 (0.94–1.12)* | 12 | 3.7 | **0.73 (0.61–0.87)*** | 0.87 (0.74–1.04)* |
| Q3 | 174 | 29.8 | | | 100 | 19.2 | | | 29 | 6.0 | | |
| Q4 | 213 | 29.1 | | | 117 | 20.9 | | | 16 | 2.8 | | |
| Q5 (most affluent) | 243 | 30.0 | | | 130 | 22.8 | | | 22 | 2.8 | | |
| **Wealth quintile-rural** | | | | | | | | | | | | |
| Q1 (least affluent) | 217 | 23.2 | ref | ref | 112 | 14.7 | ref | ref | 82 | 8.4 | ref | ref |
| Q2 | 214 | 33.7 | **1.15 (1.08–1.22)*** | **1.12 (1.06–1.19)*** | 98 | 18.3 | **1.16 (1.07–1.26)*** | **1.14 (1.04–1.24)*** | 36 | 7.7 | 0.93 (0.80–1.09)* | 1.12 (0.97–1.29)* |
| Q3 | 135 | 38.8 | | | 74 | 23.9 | | | 18 | 5.8 | | |
| Q4 | 128 | 44.8 | | | 52 | 29.3 | | | 16 | 9.0 | | |
| Q5 (most affluent) | 62 | 35.7 | | | 33 | 20.9 | | | 11 | 5.1 | | |
| **Type of residence** | | | | | | | | | | | | |
| Urban | 814 | 29.5 | ref | - | 442 | 20.9 | ref | - | 97 | 4.2 | ref | - |
| Rural | 756 | 33.2 | **1.13 (1.00–1.27)** | | 369 | 19.8 | 0.95 (0.80–1.12) | | 163 | 7.6 | **1.81 (1.28–2.55)** | |
| **Birth-related factors** | | | | | | | | | | | | |
| **Sex of the child** | | | | | | | | | | | | |
| Male | 812 | 31.1 | ref | | 418 | 19.9 | ref | | 143 | 6.1 | ref | |
| Female | 758 | 31.7 | 1.02 (0.92–1.13) | | 393 | 20.8 | 1.05 (0.90–1.22) | | 117 | 5.6 | 0.91 (0.66–1.24) | |
| **Birth order** | | | | | | | | | | | | |
| First | 585 | 38.0 | ref | ref | 300 | 24.7 | ref | ref | 97 | 7.9 | ref | ref |
| Second/later | 985 | 28.3 | **0.75 (0.67–0.83)** | **0.77 (0.69–0.86)** | 511 | 18.5 | **0.75 (0.64–0.87)** | **0.77 (0.66–0.89)** | 163 | 5.0 | **0.64 (0.47–0.87)** | **0.60 (0.45–0.81)** |
| **Perceived birth size** | | | | | | | | | | | | |
| Larger than average | 533 | 32.3 | 1.09 (0.97–1.23) | 1.1 (0.98–1.23) | 264 | 20.5 | 1.07 (0.91–1.26) | 1.07 (0.91–1.25) | 77 | 5.4 | 0.89 (0.64–1.24) | |
| Average | 830 | 29.5 | ref | ref | 429 | 19.1 | ref | ref | 154 | 6.1 | ref | |

*(Continued)*

**Table 4.** (Continued)

| | Formula (n,%) | | cPR (95% CI) | aPR (95%) | Other milk (n, %) | | cPR (95% CI) | aPR (95%) | Honey (n, %) | | cPR (95% CI) | aPR (95%) |
|---|---|---|---|---|---|---|---|---|---|---|---|---|
| Smaller than average | 207 | 39.3 | **1.33 (1.15–1.54)** | **1.27 (1.11–1.47)** | 118 | 27.4 | **1.43 (1.16–1.77)** | **1.42 (1.15–1.75)** | 29 | 6.1 | 1.00 (0.61–1.65) | |
| **Place and mode of delivery** | | | | | | | | | | | | |
| Home | 252 | 29.8 | ref | ref | 111 | 15.3 | ref | ref | 109 | 16.1 | ref | ref |
| Public facility-vaginal | 376 | 25.0 | 0.84 (0.69–1.01) | 0.92 (0.76–1.1) | 194 | 16.4 | 1.07 (0.82–1.40) | 1.18 (0.9–1.55) | 81 | 5.5 | **0.34 (0.23–0.50)** | **0.38 (0.26–0.56)** |
| Public facility-caesarean | 210 | 49.2 | **1.65 (1.36–2.01)** | **1.54 (1.27–1.87)** | 110 | 36.7 | **2.39 (1.81–3.18)** | **2.26 (1.70–3.00)** | 16 | 5.7 | **0.35 (0.17–0.71)** | **0.36 (0.17–0.74)** |
| Private facility-vaginal | 511 | 30.3 | 1.02 (0.85–1.21) | 1.14 (0.94–1.38) | 279 | 20.1 | 1.31 (1.02–1.69) | 1.43 (1.09–1.86) | 45 | 3.0 | **0.19 (0.12–0.29)** | **0.22 (0.14–0.36)** |
| Private facility-caesarean | 221 | 42.3 | **1.42 (1.16–1.72)** | **1.37 (1.12–1.68)** | 117 | 29.0 | **1.90 (1.42–2.53)** | **1.77 (1.31–2.41)** | 9 | 1.5 | **0.09 (0.04–0.21)** | **0.10 (0.04–0.23)** |
| **Antenatal card** | | | | | | | | | | | | |
| No | 173 | 34.3 | ref | | 106 | 27.4 | ref | ref | 27 | 6.5 | ref | |
| Yes | 1397 | 31.1 | 0.91 (0.77–1.07) | | 705 | 19.7 | **0.72 (0.58–0.90)** | **0.73 (0.59–0.91)** | 233 | 5.8 | 0.89 (0.54–1.47) | |
| **Immediate skin-to-skin** | | | | | | | | | | | | |
| Yes | 781 | 26.1 | ref | ref | 410 | 16.9 | ref | ref | 125 | 4.4 | ref | ref |
| No | 789 | 40.7 | **1.56 (1.4–1.73)** | **1.44 (1.29–1.60)** | 401 | 27.0 | **1.59 (1.36–1.86)** | **1.49 (1.27–1.75)** | 135 | 9.0 | **2.06 (1.51–2.81)** | **1.52 (1.11–2.09)** |

(*) = wealth quintile as continuous variable; bold figure = statistically significant; n = unweighted count; % = weighted percent.

definition of the timing of PLF (in the first three days, before regular milk flow) than in the other Southeast Asian DHS surveys (in the first three days) [44–46].

Our findings also suggest an extensive variety of PLF types among countries worldwide. Formula was the most reported PLF in Indonesia, which was similar to neighbouring Vietnam [47], but was different from Timor-Leste, a country that used to be part of Indonesia. In Timor-Leste, plain water was the predominant PLF type [48], which was similar to that in Sub-Saharan Africa [3]. Moreover, the most common PLF types were not consistent across previous Indonesian studies. Although formula was reported the most common PLF [9,12,21], honey [21,23,49], plain water [9], and sugar water [8,9] were frequently mentioned in various orders.

Additionally, other feeds such as coffee, palm sugar, dates, and coconut water were also reported [49]. Bananas or mashed bananas were among the non-liquid feeds that were given before breastmilk came in [21,23]. These feeds might be included in the 0.8% unspecified PLF in our study (Table 2) and reflect how diverse PLF is in Indonesia.

## Sociodemographic factors

Any PLF, formula, and other milk had similar risk factors, i.e. higher wealth quintiles in rural areas, most likely because formula and other milk made up more than half of the 'any PLF' group. In contrast, none of the sociodemographic factors were significantly associated with giving honey as a PLF (Table 4). Honey also had the most distinct patterns compared to any PLF and the other two PLF types as it was more prevalent among the least affluent (Q1) mothers in urban area. This could be due to the relatively higher price of formula compared to most types of honey and/or because honey is often available at home, so people don't have to specially purchase it for the baby.

Another possible explanation is that wealth quintile is also capturing a range of other differences (e.g. knowledge, expectation, family influences). However, it was not statistically significant after fitting an interaction term with residence (Table 4). This could be due to the low prevalence of honey and/or because these sociodemographic factors were only confounders for home births. For example, Titaley et al. (2010) reported that many mothers from low socio-economic status perceived going to health facilities as a costlier alternative that they often could not afford [50]. Our findings were not consistent with those observed in different settings, suggesting that PLF risk factors are not consistent across all countries. In Indonesia, maternal age and education had no association with any PLF. Meanwhile, in Sub-Saharan African countries, younger and less educated mothers had slightly higher odds of PLF [3]. Differences in PLF were also observed between mothers living in urban and rural settings in Indonesia, an association that was not statistically significant in Sub-Saharan Africa [3]. In our study, rural residence had a higher prevalence of any PLF in all wealth quintiles except the poorest quintile (Table 3), which was the opposite of Vietnam and Timor-Leste where urban residence had higher odds of any PLF [47,48]. This could be because mothers in urban areas of Indonesia are more exposed to information about the importance of exclusive breastfeeding compared to mothers in rural areas, but this explanation needs further research.

In Timor Leste [48] and Vietnam [47], the richest mothers had the highest odds of giving PLF. Meanwhile in Sub Saharan Africa, upper-middle class mothers (Q3-Q4) were the least likely to give PLF compared to Q1 and Q5 [3]. In Indonesia, similar patterns were observed but only in rural areas. Based on our observation, having formula used to be a symbol of wealth status but the trend has shifted among urban mothers as information on breastfeeding campaigns has spread, especially by social media. This trend may not have yet reached mothers in rural areas due to limited internet access.

## Birth-related factors

Formula and other milk had relatively similar birth-related risk factors to those of any PLF, but honey showed different patterns again. Sex of the child was not associated with any of the outcomes, which was in accordance with other studies [47,48], while the only risk factors shared by all outcomes were first birth and not having immediate skin-to-skin contact. In line with previous literature [3,51], these studies suggested that anxiety and/or lack of experience and infant feeding knowledge in first-time mothers may lead to difficulties in breastfeeding initiation and potentially result in PLF practice, which also appears plausible for Indonesia. Therefore, first-time mothers may require more support and education about infant feeding and the potential difficulties they may experience.

Immediate skin-to-skin contact is one of the Ten Steps to Successful Breastfeeding [18] and has been reviewed as an effective measure to promote breastfeeding [52]. Despite indistinguishable causality, our finding was also in accordance with previous studies which found that early initiation of breastfeeding was negatively associated with PLF [53] and immediate skin-to-skin contact had a positive association with breastfeeding practice [54].

The main difference between honey and all other outcomes was its association with the combination of place and mode of delivery. Caesarean deliveries (at either public or private health facilities) were risk factors for any PLF, formula, and other milk, but not for honey. In contrast, babies born at home were more likely to be given honey than those born at any health care facility, regardless of the mode of delivery.

A pooled study of 57 countries indicated that overall PLF was more prevalent among births taking place at home than those at a health facility, especially a public-owned one [20]. In our study, this association was only observed for honey. When we took mode of delivery into

account, the protective effect of health facility for any PLF was only observed for vaginal births in public-owned facilities. Vaginal delivery at a private health facility neither increased nor decreased the risk of any PLF, while caesarean deliveries at any facilities remained a consistent risk factor.

In previous studies, caesarean delivery was frequently reported as a recognised risk factor for PLF [3,47,55]. PLF after caesarean delivery could be related to failure to breastfeed resulting from postoperative pain [56], hormonal influence, higher risk of suckling failure in caesarean-delivered babies, and bonding difficulties between mother and child [57]. Birth complications that lead to caesarean may also play a role, but the available data did not allow further investigation into the underlying indication for operative delivery. Formula promotion within the health system has been acknowledged elsewhere [58–62], but our study did not have data on this issue. Further research is needed to understand why giving birth in a health facility in Indonesia did not decrease the risk of prelacteal formula and other milk, even when the mode of delivery was vaginal.

The higher prevalence of honey among home births compared to births at any health facilities may be due to inadequate information provided to women delivering at home. Many home births in Indonesia are not assisted by certified health providers [50], hence medical knowledge such as awareness of botulinum toxin may be low. Residual confounding may also explain some of the association, as home births are more frequent among women in poorer and more rural communities [50].

The statistically significant increase in any PLF, formula, and other milk for perceived smaller babies, compared to those whose mothers did not consider them small, was consistent with the existing literature [3,63]. One of possible explanations is because perceived smaller babies include preterm and very-low-birth-weight babies, the groups with more breastfeeding barriers, such as maternal stress, low breast milk supply, and separation from mothers [64–66]. Furthermore, Belachew et al [63] discussed another possibility about mothers' mistaken perception that smaller babies could benefit from additional feeds other than breastmilk. In Indonesia, perceived insufficient breastmilk is also one of the reasons for giving PLF [49]. Within this practice, it is possible that mothers who are concerned about their baby's size would believe that they need additional nutrition in order to catch-up with normal growth. This association was not seen for honey, perhaps because honey was not solely chosen for its nutritional value but also for other reasons, such as belief to bless the baby and make the baby pretty and behave sweetly in the future [23].

The protective effect of having an antenatal card was only statistically significant for any PLF and other milk, but the crude association was similar for formula and honey. Some previous studies have suggested that antenatal care (ANC) reduces the risk of PLF [3,51], but the ANC coverage was nearly 100% in this study population so it could not be analysed as a potential predictor. However, possession of an antenatal card may be an indicator of ANC quality, where better providers are perhaps the ones who give the cards. Its association with a lower PLF prevalence in Indonesia is possibly because antenatal cards contain information on the importance of EBF and warning against PLF [38]. This finding may also be related to a recent qualitative study which found that lack of information was one of the barriers to exclusive breastfeeding [10]. Although further investigation is needed to establish causality, we support the idea that educating mothers about exclusive breastfeeding is a beneficial intervention to reduce PLF practice.

## Strength and limitation of this study

This study describes the most recent picture of PLF, and the factors associated with it, in Indonesia. Most other studies of PLF in other countries have looked at risk factors of overall PLF,

without separating out the variety of feeding types within this. To our knowledge, only one other paper has compared a range of sociodemographic and birth-related factors associated with different PLF feeding types; the previous paper examined only prelacteal formula and water in Vietnam [47]. In Indonesia, this is also the first study about PLF in a nationally representative study population.

The main limitation of our study is that it is cross-sectional and therefore causation cannot be strongly inferred. However, explanatory variables were carefully selected to avoid temporal ambiguity. Potential recall bias was minimised by simple questions and a two-year recall period. The relatively low prevalence of giving honey also means that power to detect statistically significant associations was limited for this outcome. It was not possible to examine some potential reasons for giving PLF that have previously been reported in the literature [7–10,12,13,49], because the standard DHS questionnaire did not include questions about them. These include traditions, concern for newborn hunger, perceived insufficiency of breastmilk, pressure from family (husband/mother/mother in-law), advice from health providers, or failure to initiate breastfeeding within first hour. Other variables that may be related to PLF but were not analysed here included religion (data not available), delivery assistance (variable available but 95% of births were assisted by skilled birth attendants and many women were assisted by both traditional and skilled birth attendants), and marital status (variable available but 96% were married).

## Conclusion

PLF is a common practice in Indonesia, with nearly half of babies who will go on to be breastfed receiving something other than breastmilk in the first three days of life. Such PLF means that optimal breastfeeding behaviours are not being practised. As part of the governmental policy to improve the exclusive breastfeeding rate in Indonesia, this issue deserves further attention. Some of the factors identified in this study, such as first birth, caesarean delivery, and antenatal information are potentially modifiable through comprehensive education and proper anticipation from healthcare providers. Our findings will require corroboration by further research, particularly to understand specific reasons behind mothers' decisions to give PLF, and how this varies by type of PLF. However, knowing that risk factors vary across the different types of feed, our research suggests that interventions to reduce PLF may need to be targeted at different groups to reach the most vulnerable population for each PLF type. Identifying whether different factors are associated with different types of PLF is an important priority in other settings.

## Acknowledgments

Permission to use the 2017 Indonesia Demographic and Health Survey (IDHS) dataset was obtained from the Demographic and Health Surveys (DHS) Program, which was implemented by ICF and funded by the U.S. Agency for International Development (USAID). We thank ICF, National Population and Family Planning Board (BKKBN), Statistics Indonesia (BPS), and the Ministry of Health (Kemenkes) for conducting the survey, as well as all mothers who participated in the survey.

## Author Contributions

**Conceptualization:** Lhuri D. Rahmartani, Claire Carson, Maria A. Quigley.

**Data curation:** Lhuri D. Rahmartani.

**Formal analysis:** Lhuri D. Rahmartani.

**Methodology:** Lhuri D. Rahmartani, Claire Carson, Maria A. Quigley.

**Project administration:** Lhuri D. Rahmartani.

**Supervision:** Claire Carson, Maria A. Quigley.

**Writing – original draft:** Lhuri D. Rahmartani.

**Writing – review & editing:** Lhuri D. Rahmartani, Claire Carson, Maria A. Quigley.

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
