## [Decision Letter · Decision Letter 0]

13 Oct 2020

PONE-D-20-18812

Prevalence of prelacteal feeding and associated risk factors in Indonesia: evidence from the 2017 Indonesia Demographic Health Survey

PLOS ONE

Dear Dr. Rahmartani,

Thank you for submitting your manuscript to PLOS ONE. After careful consideration, we feel that it has merit but does not fully meet PLOS ONE’s publication criteria as it currently stands. Therefore, we invite you to submit a revised version of the manuscript that addresses the points raised during the review process.

I have reviewed the manuscript as the second reviewer and my comments are below. Please note that the first reviewer and I have both made comments regarding the clarity of the definition of prelacteal feeding.

We look forward to receiving your revised manuscript.

Kind regards,

Jane Anne Scott, PhD, MPH Grad Dip Dietetics, BSc

Academic Editor

PLOS ONE

Journal Requirements:

Reviewers' comments:

Reviewer's Responses to Questions

**Comments to the Author**

1. Is the manuscript technically sound, and do the data support the conclusions?

Reviewer #1: Yes

Reviewer #2: Yes

2. Has the statistical analysis been performed appropriately and rigorously? 

Reviewer #1: Yes

Reviewer #2: Yes

3. Have the authors made all data underlying the findings in their manuscript fully available?

Reviewer #1: Yes

Reviewer #2: Yes

4. Is the manuscript presented in an intelligible fashion and written in standard English?

Reviewer #1: Yes

Reviewer #2: Yes

5. Review Comments to the Author

Reviewer #1: Overall comment: This is a well presented research from the authors and to my knowledge this is the first research that used the national data set from Indonesia to investigate the factors associated with prelacteal feeding.

I have only one suggestion that the authors may consider to include in the manuscript.

Outcome variables:

I request the authors to include actual question from DHS. There is always a debate on ‘first feed’ vs ‘any food other than breast milk in the first three days’ when it comes to DHS survey. Presenting actual question would help comparability of the findings.

Reviewer #2: Definition of prelacteal feeding

As there are various definitions of PLF used by researchers, I recommend that the definition of prelacteal feeding used in this study be included in the abstract and earlier in the introduction. It is not until page 5 when the authors describe the outcome variables that it is clear that PLF includes anything other than breastmilk in the first 3 days after delivery. Various researchers have defined PLF to be ‘the first feed given’ or ‘feeding any other substance **before **first breastfeeding’, ‘foodstuff or feed that is given to a newborn infant **before **the mother has begun to breastfeed’.

While the paper for the most part is written in standard English, it would benefit from a careful final proof reading by someone for whom English is their native language. The paper includes a number of minor grammatical errors and several examples of awkward wording.

A lack of continuous page numbering made this somewhat annoying to review and comment on.

Minor corrections

Antenatal card is a noun and needs the indefinite article ‘an’ to be placed before it, i.e. had an antenatal card. There are other cases where the definite (the) or indefinite (a) article is missing from before a noun e.g. had a home birth.

Page 3, last para, line 6 should read ‘ used data from the 2017 Indonesia DHS

Page 5 line 3 should read ‘… were asked of all mothers’

Page 7 lines 14-16 the first part of this sentence is awkwardly worded. Study population

Page 9 line 4 “About 33% of the mothers had the first birth” awkward wording

Page 12 Table 2 footnote suggest rewording ‘reported as having ever breastfed’

Page 13 The text needs to stand alone from the tables. The percentages given in the first paragraph are unnecessary and not particularly useful without knowing the percentages of the comparator groups.

Page 13 When reporting the aPRs here and for other outcomes it is customary to identify the reference category. For instance, was the reference group for infant size women who perceived that their baby was average size or larger than average.

Page 15 Again, the percentages given in the first paragraph are unnecessary and not particularly useful without knowing the percentages of the comparator groups.

Page 20 Sociodemographic factors

The sentence ‘In contrast, none of the sociodemographic factors were statistically significant honey’ is awkwardly worded, suggest rewriting ”In contrast, none of the sociodemographic factors were significantly associated with giving honey as a PLF.”

Page 21 para 1 Strictly speaking your study did not highlight the diversity of PLF risk factors across countries. This wording suggests that you analysed data from a variety of countries. Instead, what you are reporting is that the associations found are not consistent across all countries.

Page 21 line 8 should read ‘higher prevalence of any PLF’

Page 21 line 10 should read ‘urban areas’ (plural)

Page 22 line 7 should read ‘immediate skin-to-skin contact is one of the Ten Steps…’ (present tense)

Page 22 line 10 should read ‘in accordance with…’

Page 24 2nd line from bottom should read ‘educating mothers’ (plural)

Page 24 line 4 suggest adding ‘within this practice.’

Reference List

Please check the reference list carefully to ensure that all journal titles are abbreviated. You may need to edit the references in your Endnote library.

Figure 1

Please check the quality of Figure 1 as it is blurry in the uploaded pdf.

6. PLOS authors have the option to publish the peer review history of their article (what does this mean?). If published, this will include your full peer review and any attached files.

Reviewer #1: **Yes: **Vishnu Khanal, PhD.

Reviewer #2: **Yes: **Jane Scott

---

## [Author Response · Author response to Decision Letter 0]

9 Nov 2020

RESPONSE TO COMMENTS FROM REVIEWER 1

Overall comment: This is a well presented research from the authors and to my knowledge this is the first research that used the national data set from Indonesia to investigate the factors associated with prelacteal feeding. 

I have only one suggestion that the authors may consider to include in the manuscript. Outcome variables: I request the authors to include actual question from DHS. There is always a debate on ‘first feed’ vs ‘any food other than breast milk in the first three days’ when it comes to DHS survey. Presenting actual question would help comparability of the findings.

Response:

We thank Reviewer 1 for raising this issue so that we can clarify. We were aware that there was a slight wording difference between the English and Indonesian versions of 2017 IDHS. In the Indonesian version, the question translates to "In the first three days after delivery, before your milk began flowing regularly, was (NAME) given anything to drink other than breast milk?", while in the English version, the question does not have the part about milk flowing regularly and it was exactly the question that we have previously included in our manuscript i.e. “In the first three days after delivery, was (NAME) given anything to drink other than breast milk?".

In our initial submission, we only included the English version because this is the standard definition used by DHS Guidelines, other DHS reports from other countries, and DHS-based studies. We also had confirmed with the DHS Team that this was the correct approach. However, after carefully considering your feedback, we agree that adding the question as worded in the Indonesian version is more accurate. Therefore, in the revised manuscript, the translated Indonesian question has been added into the methods (please refer to Manuscript with Track Changes, page 5, line 92-94). The impact of the slightly different definitions is addressed in the discussion (page 19-20, line 282-285).

RESPONSE TO COMMENTS FROM REVIEWER 2

1. Definition of prelacteal feeding. As there are various definitions of PLF used by researchers, I recommend that the definition of prelacteal feeding used in this study be included in the abstract and earlier in the introduction. It is not until page 5 when the authors describe the outcome variables that it is clear that PLF includes anything other than breastmilk in the first 3 days after delivery. Various researchers have defined PLF to be ‘the first feed given’ or ‘feeding any other substance before first breastfeeding’, ‘foodstuff or feed that is given to a newborn infant before the mother has begun to breastfeed’.

Response:

In addition to our response to the comment from Reviewer 1, we have now included the definition of prelacteal feeding (PLF) in the abstract (Manuscript with Track Changes, page 2, line 8-11) and introduction (page 3, line 27) and hope it will improve clarity.

2. While the paper for the most part is written in standard English, it would benefit from a careful final proof reading by someone for whom English is their native language. The paper includes a number of minor grammatical errors and several examples of awkward wording.

Response:

Two of the authors have English as their native language and we have revised these minor errors. 

3. A lack of continuous page numbering made this somewhat annoying to review and comment on.

Response:

We have page numbers in the manuscript, but we apologise for not including continuous line numbers. They have now been added into the manuscript. For the rest of corrections where line numbers are indicated, please refer to the Manuscript with Track Changes.

Minor corrections

1. Antenatal card is a noun and needs the indefinite article ‘an’ to be placed before it, i.e. had an antenatal card. There are other cases where the definite (the) or indefinite (a) article is missing from before a noun e.g. had a home birth.

Response:

Thank you for your detailed comments on the text. All the grammatical and typographical errors listed above have been corrected. 

2. Page 3, last para, line 6 should read ‘used data from the 2017 Indonesia DHS

Response:

Correction has been made. 

3. Page 5 line 3 should read ‘… were asked of all mothers’

Response:

Correction has been made.

4. Page 7 lines 14-16 the first part of this sentence is awkwardly worded. Study population

Response:

Correction has been made. It now says, “In this study, possession of an antenatal card was considered as a potential associated factor because it is designed as an educative intervention for pregnancy and birth preparedness. Antenatal cards contain information on breastfeeding including PLF avoidance and colostrum benefit” (line 133-136).

5. Page 9 line 4 “About 33% of the mothers had the first birth” awkward wording

Response:

Correction has been made. It now says, “One-third of the study population reported that the cohort member was their first baby” (line 170-173). 

6. Page 12 Table 2 footnote suggest rewording ‘reported as having ever breastfed’

Response:

Correction has been made.

7. Page 13 The text needs to stand alone from the tables. The percentages given in the first paragraph are unnecessary and not particularly useful without knowing the percentages of the comparator groups.

Response:

Correction has been made. It now says, “However, giving any PLF was more common among mothers who lived in rural area, had their first child, perceived their babies as smaller than average, had a caesarean delivery at public facilities, and did not have an antenatal card” (line 197-200).

8. Page 13 When reporting the aPRs here and for other outcomes it is customary to identify the reference category. For instance, was the reference group for infant size women who perceived that their baby was average size or larger than average.

Response:

Correction has been made here and elsewhere. A reference group is mentioned when reporting PRs. For example, in paragraph starting at line 201, it now says “Table 3 shows the final models after mutual adjustment for all factors. The prevalence of any PLF was higher among mothers who perceived their baby to be smaller (aPR 1.23; 95% CI1.12-1.35) than among mothers who perceived their baby to be of average size”. 

9. Page 15 Again, the percentages given in the first paragraph are unnecessary and not particularly useful without knowing the percentages of the comparator groups.

Response:

Correction has been made here. It now says, “Table 4, formula was more common among mothers who gave birth for the first time, perceived their baby to be smaller, and had a caesarean delivery (in either public or private facilities)” (line 220-223).

10. Page 20 Sociodemographic factors

The sentence ‘In contrast, none of the sociodemographic factors were statistically significant honey’ is awkwardly worded, suggest rewriting “In contrast, none of the sociodemographic factors were significantly associated with giving honey as a PLF.”

Response:

We agree that this is clearer and have amended the manuscript with the suggested text (line 301-302). 

11. Page 21 para 1 Strictly speaking your study did not highlight the diversity of PLF risk factors across countries. This wording suggests that you analysed data from a variety of countries. Instead, what you are reporting is that the associations found are not consistent across all countries.

Response:

The sentence has been reworded, and now reads “Our findings were not consistent with those observed from research in different settings, suggesting that PLF risk factors are not consistent across all countries” (line 314-316).

12. Page 21 line 8 should read ‘higher prevalence of any PLF’

Response:

Correction has been made.

13. Page 21 line 10 should read ‘urban areas’ (plural)

Response:

Correction has been made.

14. Page 22 line 7 should read ‘immediate skin-to-skin contact is one of the Ten Steps…’ (present tense)

Response:

Correction has been made.

15. Page 22 line 10 should read ‘in accordance with…’

Response:

Correction has been made.

16. Page 24 2nd line from bottom should read ‘educating mothers’ (plural)

Response:

Correction has been made.

17. Page 24 line 4 suggest adding ‘within this practice.’

Response:

Correction has been made.

18. Reference List

Please check the reference list carefully to ensure that all journal titles are abbreviated. You may need to edit the references in your Endnote library.

Response:

Thank you for highlighting this, we have checked the list and amended as needed. There are some Indonesian language texts that it is not appropriate to abbreviate, for example “Penelitian Gizi dan Makanan” in reference no.13.

19. Figure 1

Please check the quality of Figure 1 as it is blurry in the uploaded pdf.

Response:

The image for Figure 1 has been uploaded to and previewed on https://pacev2.apexcovantage.com/ and this version has been resubmitted.

---

## [Editor Report · Decision Letter 1]

16 Nov 2020

Prevalence of prelacteal feeding and associated risk factors in Indonesia: evidence from the 2017 Indonesia Demographic Health Survey

PONE-D-20-18812R1

Dear Dr. Rahmartani,

We’re pleased to inform you that your manuscript has been judged scientifically suitable for publication and will be formally accepted for publication once it meets all outstanding technical requirements.

Kind regards,

Jane Anne Scott, PhD, MPH Grad Dip Dietetics, BSc

Academic Editor

PLOS ONE

Additional Editor Comments (optional):

Please check that your references are up to date. References 16 and 53 have been published and full volume and page number details are available.
---

## [Editor Report · Acceptance letter]

20 Nov 2020

PONE-D-20-18812R1 

Prevalence of prelacteal feeding and associated risk factors in Indonesia: evidence from the 2017 Indonesia Demographic Health Survey 

Dear Dr. Rahmartani:

I'm pleased to inform you that your manuscript has been deemed suitable for publication in PLOS ONE. Congratulations! Your manuscript is now with our production department. 

Kind regards, 

on behalf of

Dr. Jane Anne Scott 

Academic Editor

PLOS ONE